# Communicating about Energy Policy in a Resource-Rich Jurisdiction during the Climate Crisis: Lessons from the People of Brisbane, Queensland, Australia

**DOI:** 10.3390/ijerph19084635

**Published:** 2022-04-12

**Authors:** Linda A. Selvey, Morris Carpenter, Mattea Lazarou, Katherine Cullerton

**Affiliations:** Faculty of Medicine, School of Public Health, The University of Queensland, Brisbane, QLD 4006, Australia; m.carpenter@uq.edu.au (M.C.); mattea.lazarou@hotmail.com (M.L.); k.cullerton@uq.edu.au (K.C.)

**Keywords:** energy sources, messaging, attitudes, qualitative interviews, Australia, environmental issues, climate change

## Abstract

There is a high degree of expert consensus that anthropogenic climate change will be catastrophic if urgent and significant measures to reduce carbon emissions are not undertaken worldwide. Australia is a world-leading exporter of coal and gas, and does not have an effective emissions reduction strategy. Though many Australians support action on climate change, this has not affected voting patterns. In this qualitative study, we aimed to explore the attitudes of Australian voters in Brisbane, Queensland, Australia towards potential environmental policies. We approached people in public spaces, and invited them to participate in interviews. Six of the thirty-five interview participants who voted for the two main political parties or were undecided voters agreed with the transition to 100% renewables and/or no new coal mines in Australia. Many thought that renewables were not reliable enough and/or the economy was too dependent on coal to make the transition. There was strong support for political leadership in order to regulate mining and pollution, and for a transition plan for fossil-fuel-dependent communities. Participants were most concerned about tangible environmental issues, such as waste and pollution, and also described needing clear solutions in order to engage with the issues. Some described feeling ‘shouted at’ by protests and messaging about climate change and environmental issues. Our findings suggest that solution-based messaging may increase levels of engagement about climate change, and that waste and pollution can be entry-points for discussions about climate change. It is important to have conversations with people about this important issue.

## 1. Introduction

The sixth assessment report of the International Panel on Climate Change (IPCC) stresses the need for urgent measures to reduce carbon emissions to avoid catastrophic climate change [1]. However, global carbon emissions continue to rise [1]. Australia is a resource-rich country with one of the world’s highest per capita emissions, and is the world’s second largest exporter of thermal coal, and the largest exporter of liquefied natural gas [2]. Australia is particularly vulnerable to the impacts of climate change given the large arid areas and highly variable rainfall [3].

There is a lot of scope in Australia to reduce emissions; yet, Australia does not have an effective emissions reduction strategy [4]. This reflects public opinion. For example, though 52% of those who voted in the 2019 Australian federal election rated climate change as an important issue, only 14% thought that it was the most important issue [5]. Therefore, climate change was unlikely to influence voting patterns of the majority of Australians. However, younger Australians indicated that they were more willing to incur a personal cost in order to facilitate climate action, consider climate change when casting their vote [5], and support renewable energy options [6].

There are two major political parties in Australia, the Liberal Party and the Australian Labor Party (ALP). The Liberal Party, a centre-right party, is in a formal coalition with another centre-right party, the National Party, but in Queensland, they have formed a single political party, the Liberal National Party (LNP). The ALP is a centre-left party. The 2019 Australian federal election was won by the LNP, a result that was not predicted by the majority of public opinion polls. The LNP had policies opposing the phasing out of fossil fuels, and actively supporting coal and gas extraction. Following this election, the losing party (ALP) reduced their commitment to combatting climate change.

There is considerable room for improvement in Australia’s climate ambition. In contrast to 153 other countries and all other major developed countries [7], Australia did not increase their emissions reduction target at the 26th Conference of the Parties to the United Nations Framework Convention on Climate Change beyond their target of 26–28% reduction on 2005 levels by 2030, committed to at the 21st Conference of the Parties in Paris. Australia committed to a target of net-zero emissions by 2050, but has not produced a road map of how to reach that target [4]. Australia has the most emissions-intensive energy system of all OECD countries except for Poland, and Australia’s energy emissions increased between 2005 and 2019 [8].

In Australia, political affiliations can be important influencers on individuals’ views about climate change. Australians’ views on political issues are closely aligned to their identification with a political party, and can be strongly influenced by party leaders [9], especially when political elites are polarised, as they have been on climate change in Australia [10]. Conversely, public opinion about an issue can influence policy, particularly when the issue is perceived to be of sufficient importance to influence voting behaviour [11]. Strong public opinion about a policy issue among swinging voters and those from a political party’s base may provide motivation for policy-makers to adopt a particular policy [12].

Queensland is Australia’s third largest state (in population) and, second to Tasmania, Queensland is the most decentralised. Its economy is heavily dependent on primary and resource industries, and this has shaped Queensland’s political culture, which is different to the political culture in southern states [13]. Though there are regional differences within Queensland, Williams [13] argues that Queensland’s political culture is characterised by five inter-related elements, including strong support for economic development (at the expense of environmental concerns), a budgetary focus on regional and rural areas, a flexible approach to policy-making based on pragmatism, and Queensland exceptionalism (we are not like other states). This political culture, which is likely due to having an economy dependent on primary and resource industries, shapes Queenslanders’ voting patterns and the issues that influence these.

Queensland was pivotal to the LNP’s election success in the 2019 Australian federal election. In the last four decades, with the exception of two federal elections, Queensland has supported the LNP on a two-party preferred basis (Australia has a preferential election system) [13]. In the 2019 Federal election, Queensland recorded the strongest swing (4.3%) towards the LNP in Australia. Though the swing was seen in all parts of Queensland, it was greatest in regional areas. In his analysis of the 2019 Australian Federal election results in Queensland, Williams concluded that the swing to the LNP was largely driven by economic concerns, including support for the proposed Adani Carmichael coal mine (a highly controversial and large coal mine that would potentially open a huge untapped coal basin), about which, the ALP was ambivalent.

There is limited research examining effective messaging around environmental issues in relation to people’s political preference, particularly in the Queensland context. Given Queensland’s strong support for economic development regardless of environmental concerns, having a greater understanding about Queenslanders’ attitudes towards environmental policy could be useful to those trying to strengthen support for strong climate action and environmental protection. The objective of this study was to explore the attitudes and positions of Australian voters living in the greater Brisbane area of Queensland towards key environmental policies. Brisbane, the capital city of Queensland, was chosen for convenience. Though Queensland’s political culture is strongest in regional areas, it is still reflected in Brisbane, particularly in outer suburbs [13]. We were particularly interested in those people who would not typically consider environmental issues in their voting decisions. Through this research, we hope to provide insight into key sentiments that can then inform environmental campaigning.

## 2. Materials and Methods

### 2.1. Data Collection

The study was undertaken across the greater Brisbane area in Queensland. Public intercept interviews were utilised to collect qualitative data exploring perceptions of, and attitudes towards, environmental policy options. This method allowed for easy access to local populations [14]. Members of the study team approached potential participants in public spaces, including sports grounds, neighbourhood parks, open malls, and footpaths near residences and businesses. These areas were chosen because they provided ample opportunity to interview participants. Any people who were alone and were standing still or sitting were approached and asked if they would like to participate in an interview. Interviews were undertaken in the Central Business District (CBD), in suburbs with a high proportion of culturally and linguistically diverse people, and in suburbs where a high proportion voted for the LNP in the 2019 federal election.

Interviews followed a semi-structured interview guide created by the project team (Appendix A). Nine open-ended questions explored perceptions of environmental issues. The first question asked participants to describe the environmental issues that they were most concerned about. Further questions asked about whether there should be no more new coal and gas mines (including a specific question about the proposed Adani mine), 100% renewable energy, and support for communities heavily dependent on coal to transition from fossil fuels. After the first ten interviews, we added a question about Australia’s Great Barrier Reef (GBR), the world’s largest coral reef system, which is situated off the coast of Queensland. We had assumed that the GBR would be volunteered as a key issue by participants. When it did not emerge in early interviews, we decided to include it as a topic prompt given that the future of the GBR has been referred to in a range of campaigns about climate change and coal mining [15]. Demographic details, including level of education and the political party voted for in the previous federal election, were also requested. Interviews were conducted by authors M. Lazarou., L. Selvey., K. Cullerton., and a postgraduate university student. After conducting an initial 16 interviews, the study team met to discuss findings and key topics to probe in subsequent interviews. No additional questions were added then.

Interviews were conducted from October to December 2019 in a range of different federal electorates and socio-demographic areas. This allowed for a diverse range of opinions to be collected. Interviews continued until data saturation had been achieved. The interviews permitted participants to discuss, reflect on, and refine their opinions over the course of the interaction [16]. Interviews ranged between 3 to 26 min in duration (average of 12 min). Individuals aged 18 years or above with a proficient level of English competency were eligible to participate. Interviews were digitally recorded and transcribed verbatim.

### 2.2. Data Analysis

We used a thematic approach to data analysis [17]. Transcripts were coded inductively using descriptive codes, with related codes being grouped into overarching themes. Transcripts were reread, with line-by-line coding undertaken to further unpack and analyse relevant themes. Team members had frequent discussions about the codes and themes to resolve any discrepancies. NVivo 12 (QSR International, Melbourne, Australia) was used to store and manage the data.

Exemplar quotes from participants are designated with an ID number, gender, age range, highest education achieved, and political party that they voted for in the most recent May 2019 federal election.

### 2.3. Ethical Approval

Ethical approval was obtained from the University of Queensland’s Human Research Ethics Committee (Approval Number: 2019002214). Informed consent was obtained from all participants.

## 3. Results

We conducted 55 intercept interviews, nine of which were excluded because the participants were not eligible to vote in Australia. A further 11 participants said that they had voted for the Greens, and were not included in this analysis because, in general, this group are very aware of environmental issues, and likely to vote for policies that support the environment. They were, therefore, not the group of interest to this study. Thirty-five participants with a wide range of ages, education level, and voting patterns remained (Table 1). Of these, only six (17%) agreed with the idea of a transition to 100% renewables and/or no new coal mines in Australia (Table 2). Six key themes were identified from the data: no alternatives to coal and gas, technology will save us and government should require it, the need for leadership for a transition from fossil fuels, I have to see it to believe it (visible and tangible environmental issues), the need for clear answers and clear solution, and bring us along with you: communicating about environmental issues.

### 3.1. No Alternatives to Coal and Gas

The belief that there was no feasible alternative to coal and gas was prominent. A number of participants (*n* = 9), most of whom expressed environmental concerns (for some, this included climate change), believed that it was not possible or sustainable to stop using coal or gas, and rely on renewables exclusively as an energy source, because renewables were insufficiently reliable. Others (*n* = 9) believed that Australia’s economy was so reliant on fossil fuel extraction that it was not viable to phase it out—because of both the impact on jobs and on export income. Some (*n* = 5) expressed both views.


*‘You know, the weather right? No one can control the weather… But, if the sun doesn′t come, you won′t get the renewable energy like the solars. Whereas the coal, at least it is an alternative.’*
ID84, male, 40–49, postgraduate university degree, LNP.

For some, the economy and jobs were their main concern. A respondent who stated they had no concerns about the environment liked the idea of 100% renewables because they are sustainable, but was concerned about job losses.


*‘Well I′ve got family working up in mines... But thn, renewable energy it, it’s like, will it work? Because the mining industry, right, if we went to renewable energy then they′d have no jobs so it would be worse off again.’*
ID63, female, 50–64, completed high school, ALP.

Some expressed uncertainty about their own views because of their environmental concerns:


*‘So I′m a little bit torn on that one [coal mining], I guess, because you need to employ people but at the same time it′s, yeah, affecting the environment.’*
ID76, Female, 40–49, completed high school, ALP.

### 3.2. Technology Will Save Us and Government Should Require It

Some participants who expressed concerns about the environment (particularly drought and water shortages) expressed a view that coal and/or gas should be mined in a way that limits harm to the environment, and that this was possible. They cited the importance of government policy requirements to limit or remediate environmental consequences of coal and gas extraction and/or combustion.

Some with these views were ambivalent about mining, recognising the benefits for jobs and the economy, while disliking the environmental impacts.


*‘I′m really pro, you know, um, the people that work in the mines having jobs and that’s kept the economy going for so long, and Australia’s economy. But I hate what it’s doing to the environment. So, I really don′t know... I think it’s probably, it’s hard. I know all the fuss about Adani at the moment. I’m sort of, I’m, again, I can see both sides. I can see the people protesting, why they don’t want it... But I also see how out where it’s going to start how the people out there are really looking forward to it. So we’ve got to find better ways to perhaps do coal mining. More environmentally friendly if there’s such a thing.’*
ID78, female, 50–64, did not complete high school, ALP.

Others were less ambivalent, believing that any environmental problems arising out of coal and gas mining can be dealt with. They did say, however, that the government should ensure that strong and enforceable environmental policies are in place so that damage is limited, and remediation is compulsory.


*‘Um, I just think we have a lot of coal. It’s obviously worked for a long time. I think that, yes, there probably are issues with, you know, the smog and the way they leave the land and things like that, but I think that all things that can be fixed... I think a lot of the things that are left behind are because the government doesn’t have enough, what would you say, enough, um, the policies in their actual approvals to make people clean up their mess.’*
ID57, female, 50–64, technical qualification, ALP.

Another respondent supported this view stating we should be more efficient in burning coal, rather than stop using it.


*‘I think it is impractical and unrealistic to assume that we’re not going to be able to mine coal. I think it’s how we go about it and what we do, to restore the environment after. And, again, more efficient use of the coal so that we get more power out of what they are taking out.’*
ID64, female, 50–64, post-graduate university degree, LNP.

Three participants who were not concerned about the GBR acknowledged that some of the reef had died, but that it could be regenerated.


*‘I heard only just recently that it’s been proven that it’s, it’s regenerative. It was just in the paper [Sunday Mail] just recently. It’s saying that parts of it are regenerating so it proves it’s not going backwards as the greenies would have us believe.’*
ID69, male 50–64, completed high school, LNP.

One respondent cited climate change as her number one environmental concern, and did not support new coal or gas mines or the pursuit of financial benefits at cost to the environment. However, she was not concerned about the GBR, because she believed it was being rebuilt.


*‘Well, they’re starting to rebuild it [the GBR] aren’t they? They have a new, I read that somewhere... Well it was beautiful to begin with. So it would be nice to see it like that again.’*
ID85, female, 30–39, technical qualification, states she does not remember who she voted for (and would prefer to vote for no political party).

### 3.3. The Need for Leadership for a Transition from Fossil Fuels

Some participants who expressed concern about climate change described the need for a longer-term plan to transform Australia’s economy away from fossil fuels. They acknowledged that without a clear transition plan, the Queensland economy’s dependence on fossil fuels means that it is not feasible either politically or economically to stop opening new coal and gas mines, even though they supported this.


*‘Today, I think our economy hangs on it [coal], so if we’re going to make that statement [no new coal mines], we better start moving pretty quickly to transfer our economy to something else. So we can’t make that statement until we’ve got a way forward… There’s probably a map… There’s a big transition for this economy. Huge transition for this economy. This entire nation is based on digging stuff out of the ground and putting it on a boat and sending it overseas pretty much. That’s what we do.’*
ID02, male, 50–64, undergraduate university degree, LNP.


*‘I agree with it [no new coal mines]. But I think to get it across politically, we need to get help to these towns. I mean there’s so many obvious things the government could do.’*
ID68, Female, 65+, postgraduate university degree, LNP.

### 3.4. I Have to See It to Believe It—Visible and Tangible Environmental Issues

When invited, participants raised a number of different environmental concerns. Quite a few raised concerns were about waste, a lack of faith that waste was being recycled, and, particularly, about plastic waste. Others expressed concern about air, land, and/or water pollution. One participant who expressed concerns about a range of environmental issues, including waste and climate change, said that waste was her major concern because it was more of a ‘day to day’ issue. She did not support stopping new coal mines, as it was important to have a balance between jobs and the environment.


*‘… generally speaking, I suppose everyone’s concerned about climate change, and... the Great Barrier Reef, wildlife, all that kind of stuff. But, to be completely honest, I think on a day-to-day basis, as just a member of the public and as part of a family, those things, probably, don’t, sort of, enter our thoughts quite as much as; it’s probably more, what we are doing day-to-day and how that’s affecting everyone.’*
ID67, female, 30–39, undergraduate university degree, LNP.

In a similar vein, there was often a lack of coherence between participants’ views on gas and coal. Though some participants were clear in their support for or opposition to both new coal and gas mines, there were those who had different views about gas compared to coal, and others who expressed a lack of knowledge about gas extraction. Some who were opposed to gas, even though they supported new coal mines, said that they did not know about the environmental impact of gas, how many gas fields we needed, or whether there was an alternative.


*‘I’m not a fan of gas. I don’t really know much about it I must admit. But it worries me what they’re extracting, they’re taking coal out of the ground too, but gas extraction is slightly different ‘cause it worries me not knowing much about it. But they take that out, what happens to all the space that they’ve extracted it from? Coal mining, it’s visible. You know, they’re digging great big holes, you can see what’s going on. But the gas is different. I’m not a fan of that at all.’*
ID78, female, 50–64, did not complete high school, ALP.

Others were more concerned about gas extraction than coal mining because of the impact of fracking. For some, this was because they heard about gas leaks or other impacts of fracking somewhere, whereas for others, it was about protecting food security or farmers losing their land. For example, this respondent was very pro-coal-mining, and did not agree with climate science, but said that we need to protect our farms.


*‘The only thing that worries me of that is the, the... going into farming land. If we, at least with the coal I believe, we are not going into areas that are farmed for coal. Whereas gas just goes anywhere. And that’s the only thing that worries me with that. I’d hate to see it, you know, becoming a gas exporter and not being able to feed ourselves.’*
ID69, male, 65+, completed high school, LNP.

Another had family who worked in coal mines and also did not think that renewable energy would work, but she was against new gas extraction because: *“… that fire that was up in Mackay where… the river, someone lit up the river and all this gas was under there. So there’s a leak.”*ID63, female, 50–64 completed high school, ALP.

### 3.5. The Need for Clear Answers and Clear Solutions

The largest proportion of participants who expressed concern about the GBR said that they did not know what could be done about it or why the reef is declining. The level of concern expressed varied between these participants. Some were very concerned:

*‘Seems quite dire; seems like... It’s really sad. It’s [GBR] obviously a natural wonder that should be protected… So certainly whatever can be done should be done. It’s such a tough one because I think everyone feels it in their heart, and I just don’t know if people know every day what every individual can do just trying to help these issues. I think we all hear about it, we all feel some sense of outrage, without necessarily knowing who we feel that towards or where we should be directing our energies.’* ID67, female, 30–39, undergraduate university degree, LNP.

This respondent, and some others, stressed the importance of individual responsibility for protecting the reef. Others ascribed responsibility to the government or ‘they’, but still did not know what could be done, or even the extent of the damage. Some thought it was the responsibility of those who raise concerns about the GBR to also propose solutions.


*‘So, obviously I want it to be sustained and kept ongoing, but, I don’t actively go looking for things on the Great Barrier Reef, so everything that I have heard, like, I haven’t heard of anything. It’s all... People saying, ‘No, we shouldn’t be doing this and that’, but I suppose we haven’t heard like what we can be doing otherwise to help it. Yeah.’*
ID55, male, 25–29, undergraduate university degree, ALP.

Some cited a range of possible causes of damage to the reef, including pollution from ships and Crown of Thorns starfish, but were not certain about the cause or what could be done to alleviate it.


*‘You cannot control the temperature of the sea, you cannot control a lot of pollution, because a lot of people go there, but you can’t stop those shipping boats or, the oil you’re getting to the water. Maybe all the effort you put onto this thing that’s only one percentage of the total impact. Before spend the money to do this thing, you gotta know what are you doing? That it works.’*
ID81, male, 30–39, postgraduate university degree, swinging voter.

Some participants stated that they knew the causes of damage to the GBR, and that there were solutions to the reef’s decline. The most cited cause was ‘traffic’ on the reef (including tourism, fishing, and ships), and those participants thought that these causes could be controlled or stopped.

In contrast, some participants who expressed concern about the GBR cited climate change and/or coal mining as the cause of the reef’s demise. They did not support coal or gas mining in principle, but some were uncertain about whether mining could be stopped due to job losses.

### 3.6. Bring Us along with You: Communicating about Environmental Issues

The interviews were carried out at the beginning of Eastern Australia’s unprecedented bushfire season in 2019/2020, and some participants referred to bushfires, drought, and/or water shortages as being their biggest environmental concern. These participants did not always describe a link between these issues and climate change, and some were supportive of the ongoing extraction of fossil fuels.

The interviews were also undertaken at a time when there were regular, visible protests in Brisbane about climate change and the proposed Adani Carmichael coal mine. Some participants mentioned climate change as an issue that they were concerned about because they had seen the protests, and that these raised the profile of climate change.


*‘I guess, you look at the amount of protests going on around the capital cities, and… hearing about it now and that obviously causes you to think about environmental issues.’*
ID41, male, 30–39, undergraduate university degree, ALP.

Most people who mentioned protests said that they understood the reasons for the protests, but not everyone agreed with some tactics, such as blocking traffic. The following participant expressed concern about climate change and the need to stop using fossil fuels.


*‘I think that there’s a debate is good.... what I don’t agree with is people gluing themselves to roads and things... So I just think they need to find another way to protest. Not kind of make it everyone else’s problem, directly by interfering with people. I think there’s a good case for it really, I think most people sit around thinking, “Well I know. We’re aware that environment affect us,” so why shove it up their noses? I think protest is good but it’s like got to go to the point.’*
ID75, male, 65+, postgraduate university degree, LNP.

Some participants who expressed concern about environmental issues spoke about the importance of educating themselves and others about them. Others described the ‘garbled’ messages in the media making it difficult to discern the truth. A couple of participants described feeling as though they were being ‘shouted’ at by climate activists without being offered any solutions.


*‘I suppose in all honesty, I’m a bit sceptical about climate change... I suppose I feel like when stuff does come on the media and people do talk about it, they’re very left-wing ideas and they shove it down your throat…. It’s rather, you’re with us or you’re not…. And when you, there’s not really anything put towards your knowledge base, its either yes or no in the media. Kind of turns me off and I’m like, “Ah well, I don’t have any idea on what’s happening”, so I’ll just... If it’s the newspaper, yep, turn the page.’*
ID55, male, 25–29, undergraduate university degree, ALP.

## 4. Discussion

Our findings provide insight into how people living in Brisbane, Queensland, Australia felt about key environmental policy issues in Australia, not long after an unexpected outcome of an Australian federal election (May 2019). When these interviews were conducted, there were frequent protests in Brisbane about the need for stronger climate action, and Eastern Australia was experiencing unprecedented bushfires. Given this context, our findings that, in general, participants aspired to a transition to renewable energy, and many were concerned about the impacts of coal and gas mining, are noteworthy. They suggest that it may be possible to convince people to support significant energy policy change, provided that this position is well-communicated and supported by policies to protect affected communities and economies. Our findings also provide the following potential novel areas for messaging to achieve cut-through.

A common theme across the different policy areas was the need for strong government leadership. This includes stricter environmental controls on mining, and to protect the GBR. In spite of a poor track record in mine rehabilitation in Queensland [18], some participants were optimistic about rehabilitation. Campaigns to require governments to hold mining companies accountable for their damage may resonate with such individuals. A number of participants identified the need for the government to develop plans to transition from fossil fuels to renewable energy. This is consistent with calls from the union movement and environmental groups, and has some support from affected communities, but has been opposed by the fossil fuel industry [19,20]. To be successful, transition plans need to be credible and provide for the economic transformation for affected communities away from fossil fuels. It is also critical that affected communities are engaged in developing transition plans [20]. In spite of climate change being a polarising issue in Australia, this suggests that proactive policies in support of climate change action may be well-supported.

Regardless of how they voted, many participants described the importance of fossil fuels for the Queensland economy and jobs. Others expressed the idea that renewables are not a reliable source of power, suggesting strong uptake of the LNP’s messaging about the unreliability of renewable energy [21]. This, combined with Queenslanders’ strong support for economic development and a belief in the importance of mining and primary industries [13], may make it challenging to convince people of the benefits and practicalities of a renewable energy future. Therefore, it may not be enough to solely provide information about solutions to the potential unreliability of renewables [22]. In Australia, attitudes towards energy sources seem to be highly polarised according to voting preferences, with political views favouring economic development being associated with support for fossil fuels [22]. On the other hand, people do tend to respond favourably to energy sources they have information about [22]. Such information may also need to clearly demonstrate how renewables can materially benefit Queenslanders, particularly those in regional areas [13].

The differences in many participants’ views about phasing out coal versus gas extraction was unexpected. In general, participants said that they knew less about gas than coal, but some thought that gas was less environmentally destructive than coal, whereas others thought that gas was worse. Those with concerns about gas were concerned about gas leakages, the impact of gas on farming communities, and on water. Other studies in Australia and internationally have found that, in general, gas is viewed as a cleaner and lower emitting energy source than coal [22,23,24]. However, when given a distinction between gas and coal seam gas, the former is viewed more favourably, and the latter is viewed less favourably [22]. The rapid growth of coal seam gas extraction in Queensland, coupled with the growth of activism targeting fracking, may have raised the awareness among some participants of the potential direct harm from fracking [25]. Coal mining has similar impacts [26], and there is an opportunity to increase awareness of the impact of coal mining on farming communities and water. Conversely, messaging about the impact of gas extraction on emissions and the local environment could be strengthened.

Australians have a very strong connection to the GBR, with 43% regarding it as the nation’s most inspiring icon [27]. High-profile risk events that impact climate icons, such as the coral bleaching that has occurred on the GBR, can elicit wide-ranging responses that amplify the climate change issue, particularly among the many international and Australian tourists who visit the reef [28]. Though there are a number of threats to the GBR and to other coral reefs around the world [29], the greatest threat to the GBR is climate change [30]. However, few participants described climate change as a threat to the GBR. This is in spite of attempts by environmental campaigns to demonstrate the link between climate change and coral reef destruction (for example, see [15]). This is consistent with the findings of Dean, Gulliver, and Wilson [29], who found that only 7.9% of surveyed Australians identified addressing climate change as a way to protect the GBR. A high proportion of our participants, when prompted, expressed concern about the GBR’s future, suggesting that understanding the impact of climate change on the GBR might encourage support for climate change solutions.

A number of participants expressed the greatest concern about more immediate, visible, and tangible environmental issues, such as waste rather, than climate change. Though it is rational to prioritise climate change action [1], climate change requires more cognitive engagement than issues such as waste, which are visible and directly attributable to current behaviour [31]. This is particularly important given the range of issues confronting people. Our study was undertaken prior to the SARS-CoV-2 pandemic, and it is possible that the pandemic will further exacerbate disengagement from less tangible environmental issues. In Australia, there has been widespread public attention on the pandemic, with limited attention to other issues, including climate change. Well-promoted solutions that are realistic, easy to conceive, and involve individual and collective actions may counteract this, and allow people to engage with the issue [31,32]. The immediacy of climate change as an issue for individuals needs to be felt. Describing the impact of climate change on others, particularly those some distance away and in different circumstances, is unlikely to be effective at engaging these people. Proposed solutions need to help people to focus on something tangible, and help them to prioritise their actions [31].

It is clear that protests raised some participants’ awareness about climate change and the impact of fossil fuels. Although raising awareness, the protests did not always change participants’ views in relation to fossil fuels. Some described feeling ‘shouted at’, and others described the irritation of having their day disrupted. It seems that these participants do not relate to those who are protesting, and therefore, do not relate to their messages. Protests can be influential in changing voters’ behaviour, particularly when the protests are occurring locally by people like them [33]. On the other hand, people who hold differing political views to those of the protesters may not be motivated by protests to change their views [33]. The mainstream media’s portrayal of the protesters as irritants or ‘terrorists’ is likely to further marginalise the protesters in the eyes of some members of the community [34]. This points to the need for environmentalists to have a human face and to initiate dialogue with people, acknowledging others’ views. Some organisations are promoting such ‘climate conversations’, which attempt to build a dialogue and shared understanding across social divides [35].

### Limitations

This qualitative study was undertaken only in Greater Brisbane, Queensland, so our findings may not be generalisable to regional Queensland areas. Further research is required in regional areas, particularly in mining regions. Our study still provides important insights into the views of Brisbane people, particularly those residing in electorates with a strong support for the LNP.

Though we actively sought out people who voted for the LNP by going to electorates held by LNP politicians, and approached people without knowing their views about the environment, it is likely that people who had an interest in environmental issues may have been more likely to agree to participate when approached. This is highlighted by the higher proportion of participants who voted for the Greens compared to the broader Brisbane population (Table 2). We removed Greens voters from our analysis to ensure we focused on those voters who may not be committed to climate action, and whose voting intentions are motivated by other concerns. This resulted in a sample of participants with a range of views about environmental issues that provided useful insights into possible future communication about climate change policies.

Our interviews were conducted prior to the SARS-CoV-2 pandemic, and as such, may not reflect current views, given the substantial disruption caused by the pandemic. This points to the importance of further research in the current context, as well as the focus testing of messaging in order to have the level of impact that the climate crisis deserves.

## 5. Conclusions

Climate change is a critical issue for humanity, but at present, this message has not reached the Australian public at the level required to support effective action. We found that participants expressed some concern about the environment, but they were not necessarily engaged with the issue of climate change. Some, regardless of who they voted for, expressed the need for stronger political leadership in this area, suggesting that they had some environmental concerns and that they expected their elected representatives to protect the environment. Our findings suggest potential messaging that may increase levels of engagement about climate change. This could include promoting solutions such as renewable energy and examples of where it works well; promoting support for communities dependent on fossil fuel extraction; and linking tangible issues, such as waste and pollution, to climate change. They also point to the importance of conversations and listening to facilitate engagement. Given Queensland’s key influence in federal election outcomes, and its strong resources sector, it is critical that environmental campaigners reach out to people living in outer Brisbane areas and in regional areas, and engage in conversations acknowledging economic concerns while progressing an agenda involving phasing out of fossil fuels.

## Figures and Tables

**Table 1 ijerph-19-04635-t001:** Demographics of the 35 participants included in the analysis.

Participants	N (%)
Age group	
18–29	5 (14.3)
30–39	9 (25.7)
40–49	4 (11.4)
50–64	10 (28.6)
65+	7 (20)
Male sex	18 (51.4)
Education	
Not completed high school	4 (11.4)
Completed high school (year 12)	8 (22.9)
TAFE/Diploma	4 (11.4)
University	19 (54.3)

**Table 2 ijerph-19-04635-t002:** Voting patterns and views on energy sources of all 46 participants eligible to vote in Australia.

Voting Preference	N	In Favour of 100% Renewables *n* (%)	No New Coal Mines *n* (%)	No New Gas Mines *n* (%)
Liberal/National party	14	4 (28.6)	3 (21.4)	6 (42.9)
Australian Labor Party	14	9 (64.3)	6 (42.9)	7 (50.0)
Greens	11	11 (100)	9 (81.8)	8 (72.7)
Other	7	3 (42.9)	1 (14.3)	1 (14.3)
TOTAL	46	27 (58.7)	19 (14.3)	22 (47.8)

## Data Availability

Data may be available upon request to the author and appropriate ethics committee approval.

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
