# Peer review of "Communicating about Energy Policy in a Resource-Rich Jurisdiction during the Climate Crisis: Lessons from the People of Brisbane, Queensland, Australia"

_ijerph, 2022, doi:10.3390/ijerph19084635_

Round 1
Reviewer 1 Report
see attachment

Reviewer 2 Report
Despite the fact it is a really interesting topic about energy policy in Australia, there are some methodological problems that make the paper not too much objective as it must be. For instance, the methodology is based on a questionnaire but it was answered only by 55 persons and 4 questionnaires were not considered so there is too much-reduced number to get an idea about the real interest or intention of these persons. A higher number of questionnaires is needed.
On the other hand, the study just does a descriptive statistics study but there are more inductive statistics that let the researchers determine, with a higher number of questionnaires, more interesting results.
Finally, the paper format is adequate and the English style is good. In consequence, the paper must be redone and part of the experiments and questionnaires, in my opinion, must be increased.
Round 2
Reviewer 2 Report
After different changes, I consider the paper adequate for publication.